# A Scoping Review on GIS Technologies Applied to Farmed Fish Health Management

**DOI:** 10.3390/ani13223525

**Published:** 2023-11-15

**Authors:** Tiziano Dorotea, Giorgia Riuzzi, Eleonora Franzago, Paulette Posen, Saraya Tavornpanich, Alessio Di Lorenzo, Laura Ferroni, Walter Martelli, Matteo Mazzucato, Grazia Soccio, Severino Segato, Nicola Ferrè

**Affiliations:** 1Istituto Zooprofilattico Sperimentale delle Venezie, 35020 Legnaro, Italy; tizianodorotea@gmail.com (T.D.); efranzago@izsvenezie.it (E.F.); mmazzucato@izsvenezie.it (M.M.); socciograzia@gmail.com (G.S.); nferre@izsvenezie.it (N.F.); 2Department of Animal Medicine, Production and Health, University of Padova, 35020 Legnaro, Italy; severino.segato@unipd.it; 3Centre for Environment, Fisheries and Aquaculture Science, Weymouth, Dorset DT4 8UB, UK; paulette.posen@cefas.gov.uk; 4Department of Aquatic Animal Health and Welfare, Norwegian Veterinary Institute, 1433 Ås, Norway; saraya.tavornpanich@vetinst.no; 5Istituto Zooprofilattico Sperimentale dell’Abruzzo e del Molise, 64100 Teramo, Italy; a.dilorenzo@izs.it; 6Istituto Zooprofilattico Sperimentale dell’Umbria e delle Marche “Togo Rosati”, 06126 Perugia, Italy; l.ferroni@izsum.it; 7Istituto Zooprofilattico Sperimentale del Piemonte, Liguria e Valle d’Aosta, 10154 Torino, Italy; walter.martelli@izsto.it

**Keywords:** farmed fish, disease, health management, GIS methods, risk mapping, risk modelling, surveillance, scoping review

## Abstract

**Simple Summary:**

In the growing aquaculture sector, more concerns have arisen regarding health management in farmed fish. Geographic information systems (GISs) dedicated to storing, collecting, retrieving, transforming, and displaying large sets of spatial data may help enhance disease control and surveillance, especially during outbreaks. To shed light on the actual implementation of GISs for health management in farmed fish in marine and freshwater environments, this scoping review analysed 54 relevant retained records from a base of over 1000 papers in the scientific literature published between 2010 and 2022. What stands out from this literature analysis is that these remote technologies are most successfully adopted in the highly valued salmonid farming industry of northern Europe and Chile. Furthermore, implementing such remote technologies and related disease mapping seemed particularly effective in controlling viral outbreaks. This review highlighted the gap between the potential of these smart GIS tools and the actual benefits accrued for disease control and surveillance when applied to farmed aquatic animals.

**Abstract:**

Finfish aquaculture, one of the fastest growing intensive sectors worldwide, is threatened by numerous transmissible diseases that may have devastating impacts on its economic sustainability. This review (2010–2022) used a PRISMA extension for scoping reviews and a text mining approach to explore the extent to which geographical information systems (**GIS**) are used in farmed fish health management and to unveil the main GIS technologies, databases, and functions used to update the spatiotemporal data underpinning risk and predictive models in aquatic surveillance programmes. After filtering for eligibility criteria, the literature search provided 54 records, highlighting the limited use of GIS technologies for disease prevention and control, as well as the prevalence of GIS application in marine salmonid farming, especially for viruses and parasitic diseases typically associated with these species. The text mining generated five main research areas, underlining a limited range of investigated species, rearing environments, and diseases, as well as highlighting the lack of GIS-based methodologies at the core of such publications. This scoping review provides a source of information for future more detailed literature analyses and outcomes to support the development of geospatial disease spread models and expand in-field GIS technologies for the prevention and mitigation of fish disease epidemics.

## 1. Introduction

Aquaculture is one of the fastest growing food production sectors in the world [1], and its production rate is expected to double over the next two decades [2] due to the continuously increasing demand for aquatic products, which is not supported by the number of wild species captures [3]. However, aquaculture is vulnerable to many infectious and non-infectious disease outbreaks that can result in a partial or total loss of production [4]. Transmissible diseases and the measures required to control the outbreaks may have especially devastating impacts on animal and public health and the economy. For modern intensive aquaculture systems, the increased global movement of aquatic animals and final products, as well as several new sources of anthropogenic stress to aquatic ecosystems, have led to the emergence and re-emergence of finfish diseases (e.g., viruses and sea lice) and health issues, which is one of the most significant risks for the sustainability of the sector [5,6].

Surveillance is paramount for controlling animal diseases transmitted via water, animals, animal products, biological materials, and other vectors, such as wild populations, where diseases that do not affect wild hosts can be problematic if transferred to an aquaculture setting [7]. To ensure the best use of surveillance and disease response resources, farming and environmental data should be collected, shared, and used as effectively and efficiently as possible. Therefore, successful disease control and prevention strategies are highly dependent on knowledge of the geographical location and extent of disease occurrence and the environmental characteristics of susceptible species, among other factors [8].

In this context, many veterinary institutions and national fisheries have been suggesting a set of primary strategies to prevent, control, and reduce infectious outbreaks. This is especially true in salmonid farming, where great emphasis is on preventative measures, disease surveillance, control, and research to reduce the incidence of animal diseases and minimise the impact of their outbreaks [9]. As shown in these strategies, adopting current protocols on key infectious pathways for infectious diseases notably enhances disease control prospects, allowing for targeted surveillance and intervention, such as eradication programmes involving the implementation of restricted zones and the restriction of movements. To this aim, mathematical and statistical modelling approaches have been suggested and used successfully to disentangle infection pathways associated with intensive marine finfish farming [10,11] and risk factors for disease transmission in wild populations [12,13].

To support the development of risk-based monitoring and surveillance systems to ensure optimal animal health management in the veterinary domain, geographic information system (**GIS**) technologies have proved to be reliable in modelling the spatiotemporal risk of exposure to disease, assessing the probability of disease transmission, and controlling the spread of outbreaks [14]. A GIS is an information system dedicated to storing, collecting, retrieving, transforming, displaying, assembling, and manipulating large sets of geographically referenced data, with a range of functions and methods for analysis of spatial data. With its ability to handle diverse types of data (e.g., point locations, linear features, polygonal features, remote sensing data, and attributes that vary in space and time), the GIS has become an important device to support biosecurity and epidemiology, providing insights for hypothesis generation of potential risk factors for disease spread, related harmful impacts on reared finfish, and the health status of wild populations. Indeed, within the one-health concept, GISs have shown their capability to integrate information from different sectors and promote collaboration using a sustainable, holistic, and transdisciplinary approach [15].

However, GISs have not been fully exploited for their ability to integrate multi-sectoral analyses and risk modelling in the interlinked areas of aquaculture and wild fish population dynamics, nor disease surveillance and control management [16]. This includes several reasons, such as both spatial and temporal limitations in GIS data collection and resolution to underpin an epidemiological protocol for rapid response during disease outbreaks. Moreover, remote sensing techniques could offer an accessible solution for mapping large, complex, and difficult-to-access aquaculture sites to enhance the prevention and control of farmed fish diseases. However, the potential of GISs, coupled with several epidemiological models, has been recently presented as an ideal solution for analysing relationships between prevalence and incidence as disease outcomes, combined with calculated risk factors as explanatory variables, to construct risk maps [9,10,17,18]. An acoustic-telemetry-based system linked with a GIS was exploited to evaluate the interactions between escapee-farmed and wild fish to help monitor the effects of interbreeding, transmission, and resource competition [19]. In another study, watershed modelling via the ArcView program identified the relationships between road density, or ecological traits, and the severity and prevalence of infections in caged sentinel rainbow trout [20].

This scoping review aims to provide an overview of the extent of the literature surrounding the current use of GISs globally to support veterinary services in farmed fish health management in freshwater, marine, and transitional environments. Indeed, scoping reviews are deemed to summarise a wide range of published evidence to understand the amplitude of a discipline, evaluate the extent and nature of research activity, and display gaps in the literature. Therefore, this explorative review study summarises original research articles that report GIS applications, databases, and functions useful for risk and predictive modelling in fish health management. We also hope to identify and characterise spatiotemporal and epidemiological data frequently collected using GIS methodologies, potentially suitable in aquatic surveillance programmes to enhance the biosecurity of the environment as a whole ecosystem. To the best of our knowledge, this is the first review investigating such specific topics, especially in the niche of GIS implementation for farmed fish health management, that provides further novelty by applying text mining and research area modelling on the final selected records.

## 2. Materials and Methods

### 2.1. Protocol and Eligibility Criteria

The protocol used for this scoping review was drafted using the preferred reporting items for systematic reviews and meta-analysis (**PRISMA**) extension for scoping reviews (**PRISMA-ScR**), including only primary research studies published between 2010 and 2022, available in full text, and written in English [21]. Their main domain had to be the aquaculture sector, but articles on wild fish were included only if close interactions with the farmed population clearly caused health implications for the wild counterpart. Within this domain, studies were included if all the following topic areas were somehow included among the objectives of the experimental design:Aquatic environments (freshwater, marine, and transitional environments);Diseases (listed by the World Organization for Animal Health (**WOAH**) [22], i.e., diseases with major economic impact);Epidemiological issues (animal health, disease surveillance, monitoring plans, and disease response);GIS issues (methodologies, applications, type of analysis, and software).

Sources of grey literature were excluded from the search, as we considered that their inclusion in this scoping review might have generated results that are difficult to compare with those from peer-reviewed articles.

### 2.2. Information Sources and Literature Search

The following databases were searched to identify potential references for inclusion: Ebsco Essential, Web of Science, and Scopus^®^. The search, based on keywords, was formatted accordingly for all three databases using the following list of words or abbreviations: GIS or spatial or cartography/cartographic or map or geographic/geographical or aquaculture or fish or wild fish or disease or surveillance or epidemiology/epidemiological or monitoring or disease response. A further specific search strategy based on the following strings was implemented on the titles, abstracts, and keywords:

(((gis) OR (spatial) OR (cartograph*) OR (map*) OR (geographic*)) AND (aquaculture) AND ((fish AND disease*)) AND ((surveillance) OR (epidemiolog*) OR (monitoring) OR (disease AND response))) AND (((gis) OR (spatial) OR (cartograph*) OR (map*) OR (geographic*)) AND ((wild AND fish) AND (aquaculture))).

### 2.3. Selection of Sources of Evidence

The screening and final selection of eligible references were performed over two subsequent phases. The first screening phase was conducted by two authors to verify whether the titles and abstracts fell within the topics of this scoping review. This first screening phase was based on the following questions:Does the title/abstract make reference to GIS technologies/applications, spatial analysis, or mapping procedures?Does the title/abstract make reference to the type of aquaculture farms or wild fish populations of interest for this review?

For submission to the second screening phase, both questions had to be answered positively by both screening authors, and disagreements were resolved by a third author. Then, the second screening phase was conducted by the first two authors, who analysed the full-text articles to evaluate the objectives and methods of the studies. In this second screening phase, the questions to be answered were as follows:Does the study describe an application of GIS technology within the aquaculture sector alone, in aquaculture sector interactions with the wild fish population, or in environmental issues that can affect farmed fish health?Does the study describe the support activities of veterinary services or farmed fish health services?Does the study describe surveillance activities or epidemiological investigation/analysis in the aquaculture sector alone or in aquaculture sector interactions with the wild fish population?

At least one question had to be answered positively by both screening authors to match the objectives and methods of this scoping review and, hence, to be submitted for final evaluation. During this phase, articles with unclear purposes or with experimental methods not comprehensive enough to answer the above-mentioned questions were rejected. Disagreements between the two screening authors were resolved by the same third author.

### 2.4. Data Charting Process and Data Items

The data charting process was carried out using a data-charting form (Excel spreadsheet) previously developed for the extraction of data from the selected studies. Subsequently, the following details, information, and data were extracted from all the selected references:Bibliographic (DOI, title, authors, journal, year of publication, and keywords);Country/location (where the studies were conducted or to which they referred);Purpose of the study (e.g., disease surveillance, health status/outbreak investigation, or disease management support);Demographic (species investigated, diseases, type of environment, and population);Epidemiological and surveillance (farm, outbreak and environment, fish movement, and control methods);GIS application: purposes of GIS use (to visualise the study area, to visualize the analytical model results, to implement GIS methods);GIS methodology: operations and functions to elaborate data (centroid, Euclidean distance, seaway distance, buffer, point-in-polygon, data map algebra);Type of analysis: visualisation (data distribution, overlay), geostatistics estimation (kernel, kriging), clusterisation (k-function, Moran’s I, Knox test, nearest neighbour);Modelling: statistical or mathematical models (hydrodynamic model, particle-tracking model, scan statistic, basic reproduction number R_0_, logistic regression);GIS program: software to implement GIS applications and modelling (ArcGIS, QGIS, Microsoft Excel, R package, CrimeStat, SaTScan).

Based on the extracted data, a descriptive analysis of the selected references was performed, and the main study characteristics were described.

### 2.5. Text Mining and Research Area Modelling

Text mining and research area modelling were performed using the R package (v4.0.2; R Core Team 2022). A document-term matrix was calculated. A term frequency-inverse document frequency (**TF-IDF**) technique was applied to weight the number of times a word stem appeared across titles, abstracts, and keywords of the selected records. A research area modelling analysis was carried out using latent Dirichlet allocation (**LDA**) to generate research theme representations according to the frequency distribution of word stems within titles, abstracts, and keywords via an iterative probabilistic process using a Gibbs sampling option of the research area models package in R. The five generated research themes based on the ten most probable word stems were presented as an unstructured set of word stems using histograms, where every bar representing each word stem is proportional to the probability (beta value) of finding it in a theme [23].

### 2.6. Synthesis of Results

The data items described in Section 2.4 are displayed in the charting form provided as Appendix A. Figure 1 describes the workflow for selecting the eligible references. Subsequently, the extracted data were synthesised using a combination of narrative text and graphical methods. A summary classification of the references according to the study characteristics (environment, GIS methods or techniques, objectives) is shown in Table 1. Yearly and geographical distributions of the included records are shown in Figure 2 in panels (a) and (b), respectively. The proportions of investigated species within country and the pathological agents within species and environment are shown in Figure 3, Figure 4 and Figure 5, respectively. The histograms in Figure 6 display the five research themes as the output of the text mining (LDA procedure) analysis of word stems applied to titles, abstracts, and keywords of the 54 selected records. A cloud representation of the most frequently used word stems according to the TF-IDF ponderation system is reported in the Appendix A (Figure 7). 

## 3. Results

### 3.1. Selection of Sources of Evidence

The flow of the selection of the references via the screening phases is described in Figure 1. The search of the three databases returned 1140 initial records. After removing duplicates, 879 single records were submitted to the first screening phase on titles and abstracts, which allowed for the exclusion of 805 mainly because they were off topic. Therefore, 74 records were submitted to the second screening phase on full texts, during which 20 records were excluded. A total of 54 records were retained for inclusion in this scoping review.

### 3.2. Characteristics and Results of Sources of Evidence

The yearly and geographical distribution of the included and analysed references are displayed in panels (a) and (b) of Figure 2, respectively. In Table 1, the selected references are classified according to the environment, GIS methods or techniques, and objectives of the studies. As required by the eligibility criteria, all selected studies involved GIS technologies applied to farmed stock or wild populations of finfish when interactions between the two populations occurred. Analysis of the charting form (Appendix A) and the summary classification (Table 1) highlighted that the main objectives of the selected references were related to risk factor analyses (*n* = 18), epidemiological examinations (*n* = 9), descriptive investigations (*n* = 10), disease spread (*n* = 7), environmental issues (*n* = 6), and surveillance and biosecurity plans (*n* = 4).

Regarding GIS applications, all papers used geographical data in one or more images to represent the study area and/or the results of the study. GISs have been used exclusively to visualise the investigated areas in five papers, and, in 14 papers, to represent the results of the study graphically. Additionally, in 18 studies, both the investigated areas and results were supported with a map and use of GIS software. Spatial data manipulations, such as buffering or distance measurement, were performed in 17 papers. Across the studies, the most commonly applied methodologies were as follows: seaway distance calculation (*n* = 19), buffering (*n* = 16), Euclidean distance calculation (*n* = 10), nearest neighbour search (*n* = 5), point in polygon (*n* = 4), map algebra (*n* = 5), and centroid calculation (*n* = 4). Less frequently used methodologies included the following: overlay analysis, area calculation, Haversine distance, spatial filtering, classification, basic cartography, and generalisation. Regarding the algorithm component, it was possible to identify a modelling approach in most (*n* = 42) papers. Image representations and spatial analysis were performed using the R-software (*n* = 7) or ArcGIS (*n* = 7) alone or in combination with other software, such as QGIS, GIMP, GoogleEarth, WinBugs, ERDAS, Satscan, and STATA. Less frequently used software packages were as follows: Acolite, CrimeStat, Python, Ecosim, Gedis, SAS, ARC CALC, and OpenGeoDa. However, 14 references did not explicitly indicate the software used to represent the data or to perform the analysis. The most frequently studied fish species was Atlantic salmon (*n* = 21, exclusively; *n* = 11, together with trout species); a few other references concerned a single species, such as rainbow trout (*n* = 2), sea trout (*n* = 2), Atlantic cod (*n* = 1), European sea bream (*n* = 1), and topmouth gudgeon (*n* = 1). The remaining 15 references included more species or referred to fish generically. The most frequent fish diseases or issues were sea lice, infectious salmon anaemia virus (**ISAV**), and virus-induced pancreas disease (*n* = 10, *n* = 8, and *n* = 6, respectively). An additional 19 publications referred to other, less common, specific viral, bacterial, or parasitic diseases or a generic risk of disease. Meanwhile, 11 studies focused on the health risks from interactions between farmed and wild populations within a restricted intensive farming area in both marine and freshwater environments. Most (*n* = 39) of the studies were performed exclusively on marine environments, while fewer papers were based on freshwater environments (*n* = 9) or a mix of marine and/or freshwater and/or transitional environments (*n* = 6). Most papers used coastline background information in the maps (*n* = 34), along with other data, such as bathymetry, sea regions, sea depth, ports, and harbours. Other background mapping information included administrative division, type of seabed, hydrography, rivers, inland facilities, sea currents, riparian zones, and coral reefs (Appendix A). The main attributes of the collected data items are reported as proportions of investigated species within a country (Figure 3), proportions of pathological agents within species (Figure 4), and proportions of pathological agents within the environment (Figure 5).

### 3.3. Characteristics and Analysis of the Research Clusters

In the text mining analysis, the pre-processing of the data produced 1928 stems that were retained from the 54 selected records. According to their TF-IDF, the 10 most relevant word stems were lice (TF-IDF = 1.04), sav (0.99), isav (0.97), sea (0.91), coral (0.66), salmon (0.63), reef (0.62), chile (0.61), cage (0.60), and temperature (0.55). As reported in Figure 6, the five generated research areas were tentatively assigned the following themes: (1) marine intensive fish farming and environment, (2) outbreak viruses in salmon marine farming, (3) risk factors for trout diseases, (4) disease epidemiology and modelling in salmon farming, and (5) fish rearing management. The same informative data are exploited in the cloud representation available in Figure 7, where the probability of finding the word stems is proportional to their font size.

## 4. Discussion

This present scoping review aimed to investigate the current implementation of GIS technologies in original research papers on farmed and wild fish when interactions between reared stock and surrounding free-living aquatic populations occurred. Across the analysed references, it was possible to verify the actual use of GIS, mainly coupled with spatial and network-based risk models and assessments, aiming to enhance insights into disease spread dynamics and the magnitude and duration of the related epidemiological phenomena. Furthermore, scenario modelling of spatiotemporal data within a GIS was seen as a useful approach for the development of risk-based monitoring and surveillance systems to support the implementation of preventative measures for veterinary and health services in aquatic animal health management.

The restricted number of included papers (Figure 1) confirms the limited use of GIS technologies for fish health management purposes regarding active surveillance programmes and a predictive multifactor modelling tool to prevent and control disease outbreaks. Analysis of the yearly distribution of publications indicated that studies were published fairly evenly across the entire period considered, with high and low points in 2015 and 2017, respectively. This further confirms the absence of a particular focus within the current research community on exploring the use of advanced GIS technology to enhance preventative strategies in aquatic epidemiology and risk analysis. This is contrary to the advances made in developing effective veterinary prevention and control measures associated with emerging and resurging vector-borne diseases in humans [62,63] and parasitic animal diseases [64]. Furthermore, remote sensing imagery was used to map environmental attributes to build a multi-agent simulation of the spatial and temporal transmission dynamics of foot-and-mouth disease between livestock and wildlife [65], and a Kernel smoothing technique applied in ArcView software was used to plot the distribution of bovine-spongiform-encephalopathy-positive holdings across Great Britain, using factors influencing the quantity of recycled infectious material to which bovines were exposed [66].

The meta-analysis of the selected references indicated that the current use of GIS technologies is clustered in terms of country, species, disease, and type of environment of application. The vast majority of the studies were conducted on the European and American continents, with Norwegian and Chilean research (Figure 2), and associated aquatic product stakeholders, being the predominant users of these tools for improving disease control and increasing production performance [67]. Moreover, a smaller number of similar studies were performed on sea farming areas of the UK and the Mediterranean, while very few studies were conducted on the other continents. Domestic salmonid species were the main target of the literature records selected using the adopted method, likely due to the high investment and advanced GIS knowledge required to develop and implement a stochastic, spatially explicit modelling framework to simulate the waterborne spread of infectious pathogens. This assumption is supported by the fact that the small number of other finfish species involved were also high-value marketable stock (e.g., cod, tuna, and sea bream).

The powerful capabilities of GIS have been demonstrated in coastal and marine resource management (water depth, buffers around existing activities, vessel tracking, etc.) [7]. However, it should be noted that over time, an interest in investigating other GIS applications has emerged, such as estimating the spatial distribution of ecosystem services in wide transitional areas of high ecological value [68], water/land use risk assessments at multiple spatial scales [20], and prioritising habitat restoration efforts for inland wild trout [69]. Moreover, GIS data collection allowed for model elaboration in order to have the efficient use of natural resources (e.g., optimal temperature range) targeted at higher-risk animals and geographic areas for early disease detection, which represented a reliable bioinformatic platform for future refinement [50]. This study investigated the change in distribution of high-risk areas for both exotic and endemic cyprinid fish diseases under different environmentally changing scenarios, making a worthwhile contribution to the surveillance of exotic pathogens.

Considering the associations between results on farming locations and environments, as well as diseases investigated, marine sites dedicated to host cages for the production of salmonids seemed to be an ideal setting for the application of GIS tools [18,44]. Indeed, countries such as Norway, Chile, and the UK have long coastlines that, regardless of indentation, are an ecological continuum where pathogens can spread quickly. Viral diseases are the most represented pathologies, especially in spatiotemporal studies considering, for instance, interactions between non-native escapee fish and the wild population [19]; risk mapping for disease prevention [13,46]; spatial conflict between marine cage farms and reefs, and associated biological implications [70]; and the assessment of the environmental impact of cage spatial clustering using geodata as a GIS information source [61]. Furthermore, sea lice, another typical pathological agent of marine environments (defined as a “burden on sea trout” by Middlemas et al. [37]), were often found to be the target pathological agent in the included studies. The application of geodata permitted the development of a multivariate spatial (Poisson) model to cluster the area of elevated sea lice infestation [38] and the assessment of efficiency of pyrethroid use at the farm level via spatiotemporal clustering of sea lice abundance before and after treatment [34].

It should also be noted that, through this meta-analysis of the literature, we noted that innovative raster techniques (e.g., map algebra, zonal statistics, and interpolation) seemed to be used to a limited extent in fish health management, while rasterised spatial data and their integration with remotely sensed data can be exploited to fulfil the vector-based solutions and develop a comprehensive approach in spatial data analysis. This is especially suitable in aquaculture, where the physical medium is a spatial environment in which natural phenomena change continuously. Consideration of the reasons behind the reduced application of innovative GIS technologies in current aquatic veterinary medicine, especially when raster-based approaches are preferred over vector-based approaches, is beyond the scope of this review. Nonetheless, we should recognise that improvements in the quality and resolution of satellite raster data and advances in computing processing performance are leading to rapid progress in these applications.

The use of GIS as a primary operational tool for the surveillance, monitoring, and early detection of changes in the incidence of infectious diseases is explored and described in a very limited number of papers, e.g., Mardones et al., who applied such tools to assess the time–space clustering of disease outbreaks (i.e., Knox test) [30]. Based on space–time interactions in fish, the study estimated the physical distance that would reduce the risk of an infected fish from a single sea cage shedding and transmitting infectious agents (e.g., viruses) to adjacent cages. From an epidemiological perspective, the Mardones study provides the maximum transmission distance at which viral transmission may occur between cages (promoting the within-farm spread of a pathogen), thus improving the capability of farmers and veterinarians to prevent and mitigate their impact. In the literature, several models were evaluated to find the best fit to assess the relationship between the prevalence and incidence index as the outcome variable and the calculated risk factors, as explicative variables. For instance, Guerrero-Cabrera et al. compared ordinary Kriging and inverse distance weighted interpolation (**IDW**) as interpolation methods to construct risk maps for bacteriosis, trichodiniasis, and girodactylosis in tilapia and rainbow trout farms over three years, highlighting that IDW was more adequate as a prediction method [12]. A logistic regression was used to model the likelihood of an outbreak of an infectious disease causing losses to the salmonid farming (i.e., heart and skeletal muscle inflammation, **HSMI**), including many risk factors, such as cohort indexes, infection pressure, and a polynomial regression line of the geo-index position of a group of marine farms located along the Norwegian coastline [32]. A further example is a study by Stene et al., where ocean model systems were used to couple physical and biological processes, as well as to identify how farms were linked by sea currents, in order to predict the risk of disease transmission via water so that timely preventative actions could be implemented [57]. Similarly, observations from automatic vessel tracking systems (automatic identification system, **AIS**), which provide detailed positioning data in real time, were used to develop a model to simulate pancreatic disease spread in marine farms arising from vessel movements and associated farm seaway distance [18]. Another application was the assessment of spatial and temporal patterns in infection dynamics through an analysis of outbreak data, involving a mechanistic model using a GIS application based on PBSmapping software, to manage emerging infectious diseases in wildlife populations, as described by Peacock et al. [35].

Despite the numerous potential implementations of GIS for farmed fish health management, this literature analysis also showcased that the stakeholders operating in the sector lack both practical and theoretical knowledge and familiarisation with the available GIS-based tools and their applications. GIS training and access to standardised sources of information should be boosted, together with the consistent and timely collection of useful surveillance data through adequate monitoring infrastructures (both spatial and temporal). As an example, management and prevention of disease introduction via water or other transboundary vectors rely on surveillance systems for data collection, recording, and control, especially to ensure the sustainability and social acceptance of salmon aquaculture [31].

The synergistic contribution given by integrating text mining within a scoping review proved to be effective in grasping the current multidimensional usage of GIS technologies applied for disease surveillance and response in farmed fish health management. As shown in Figure 6, which presents the text mining results from the aquatic-related text corpus (i.e., titles, abstracts, and keywords) within the topics and objectives covered in the 54 selected records, the ten most probable words per research area (research area of text mining) highlighted, once again, the limited range of investigated species, rearing environments, and diseases, as well as the lack of GIS-based methodologies at the core of such publications. Indeed, the most remarkable output displayed in the text mining analysis is that the use of GIS technologies did not emerge explicitly as the main objective of the publications. The literature mining identified and brought to light only word stems that are linked with but do not belong exclusively to GIS technologies (i.e., spatial or model). This was also confirmed through in-depth reading of the full-text articles, which provided evidence that GIS and related technologies are still secondary among the included references, as these papers do not describe their application but only mention their use as a means to a further objective (Figure 7). The potential of GIS as an effective tool for mapping and performing spatial data analysis is often underestimated. In the records analysed, only a few available analytical methods and data modelling knowledge areas are exploited with only basic analytical operations and geometric measures. GIS technologies were not maximised, and remotely sensed data were rarely used in the selected studies, despite being solutions that could be applied successfully in surveillance and disease response in aquatic environments. On the other hand, Rodríguez-Benito et al. demonstrated that making use of Copernicus Sentinel-2 and Sentinel-3 data is a powerful way to optimise the decision-making process in monitoring natural risk factors (i.e., algal blooms) that can negatively impact farmed finfish production performance and health [47]. The application of “next-generation optical sensor” satellites (spectral and spatial acquisition devices), instead of more traditional in situ monitoring, reduces time and cost when estimating risks posed by a natural phenomenon, such as elevated chlorophyll concentration. Long-period (15 years) satellite-derived remote sensing reflectance data were implemented to develop a remote sensing algorithm to predict harmful algal blooms in the Red Sea, which helped policy-makers to develop decision-making and risk mitigation processes dedicated to dealing with environmental and aquaculture conflict management [71].

This scoping review unveiled unrealised opportunities for the use of GIS technologies in supporting fish health management. There are only few studies whose output is the investigation of large-scale spatial distributions and long-term temporal (interannual and/or seasonal) evolution of fish diseases to perform models for outbreak simulation in real time to assess the probability of disease transmission. Such investigations would be essential for developing fish health management strategies based on robust surveillance plans and applications dedicated to integrating farm biosecurity. Moreover, this may lead to the promotion of both sustainable aquaculture operations and other regional businesses that share the aquatic environments (i.e., fisheries, tourism).

## 5. Conclusions

This scoping review analysed and described a literature text corpus surrounding the actual implementation of GIS technologies over the last decade to support farmed fish health management, including suggestions for risk assessment and predictive modelling techniques that could enhance epidemiological and surveillance programmes. The limited number of records included in the text corpus is due to the intentionally strict selection of keywords and eligibility criteria applied during the literature search, as well as the exclusion of grey technical literature that may report applications of GIS technologies not considered in the bibliographical repositories used. The main conclusion is that intensive aquaculture systems and highly valued food products are the discriminant factors justifying the use of GISs for health monitoring and enhancing the effectiveness of epidemiological models, especially for transmissible diseases that can undermine production efficiency in aquaculture. The consistently low publication rate confirms that the high investment and advanced knowledge required for applying GIS technologies puts them beyond reach for use in other low-input aquaculture systems. Both the PRISMA approach and the text mining analysis highlighted that GIS technologies are mostly employed in research activities for the visual delineation of disease areas, as well as tracking interactions between farmed and wild fish populations or farmed fish and environmental biota, rather than for implementing risk factor analysis and modelling disease in actual surveillance programmes. A gap appeared to emerge in operative research promoting GIS implementation to interlink farmed fish health with the surrounding ecosystem, making the most of up-to-date environmental, ecological, and epidemiological inputs (e.g., from in situ monitoring and remote sensing at a range of spatiotemporal scales) to map, prevent, and control outbreaks and the spread of disease.

According to this literature analysis, the application of GIS to support surveillance and disease response in aquaculture still seems focused on mapping study areas, outbreak distributions, and plotting outputs of statistical or mathematical modelling. Despite the potential of GIS-based tools to develop analytical methods, data models, and cartographic and visualisation solutions, the aquaculture stakeholders appear to need training and familiarisation with these digital technologies and statistical modelling. Hence, the identification of barriers limiting GIS exploitation in surveillance and disease response can help establish theoretical knowledge and technical expertise that should be developed and disseminated for fish rearing. Therefore, future research design should employ GIS tools and techniques to facilitate the update and implementation of large operational and randomised geospatial datasets and to enhance the understanding of connectivity and spread of infectious agents during fish disease epidemics. This challenge could be achieved by enhancing network-based and geostatistical disease spread models with associated environmental data to investigate transmission dynamics and the magnitude and duration of epidemics more realistically. This would enable scenario modelling to underpin the development of risk-based monitoring and robust surveillance systems for the timely notification of disease outbreaks in support of farmed fish health management.

## Figures and Tables

**Figure 1 animals-13-03525-f001:**
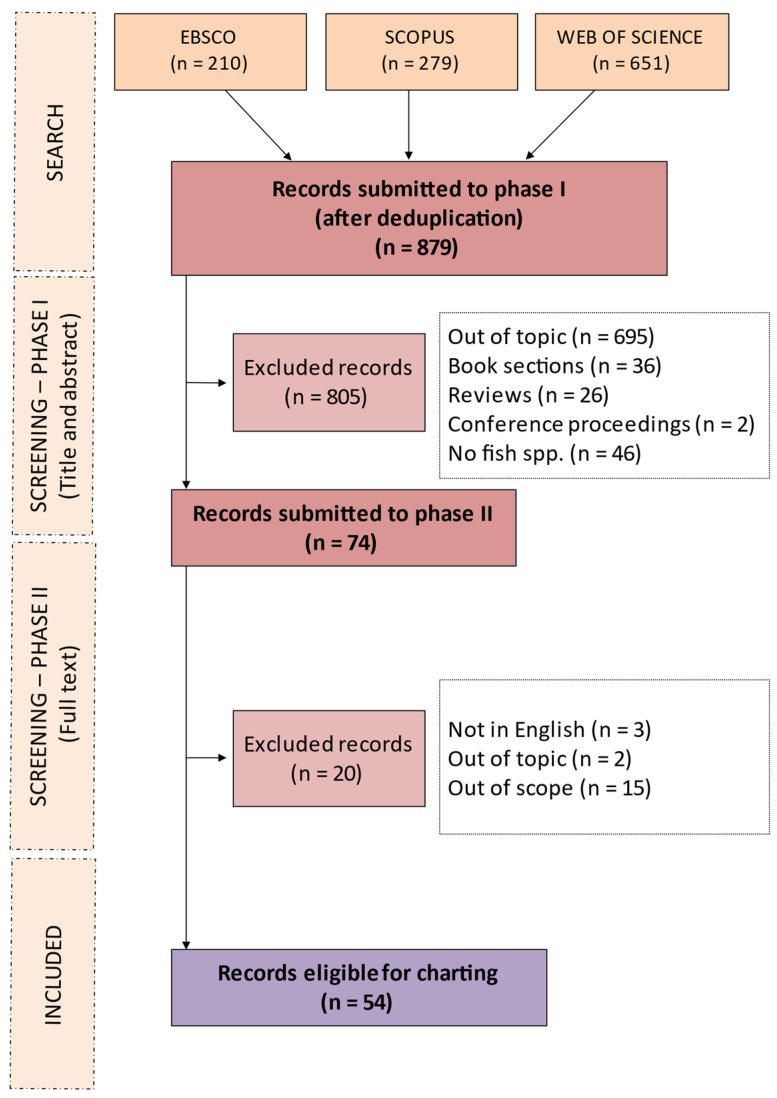
Preferred reporting items for systematic reviews and meta-analyses for scoping review (PRISMA-ScR) workflow used for the literature search surrounding GIS implementation for farmed fish health management.

**Figure 2 animals-13-03525-f002:**
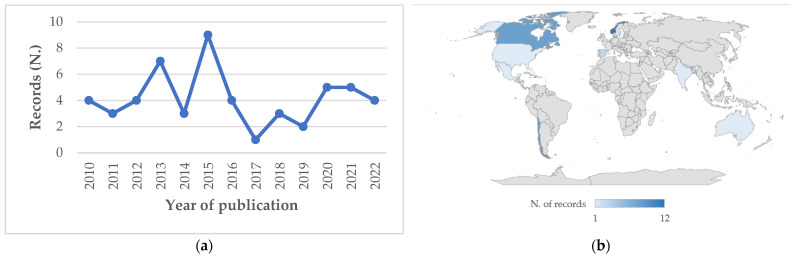
Yearly (**a**) and geographical (**b**) distributions of the 54 selected records.

**Figure 3 animals-13-03525-f003:**
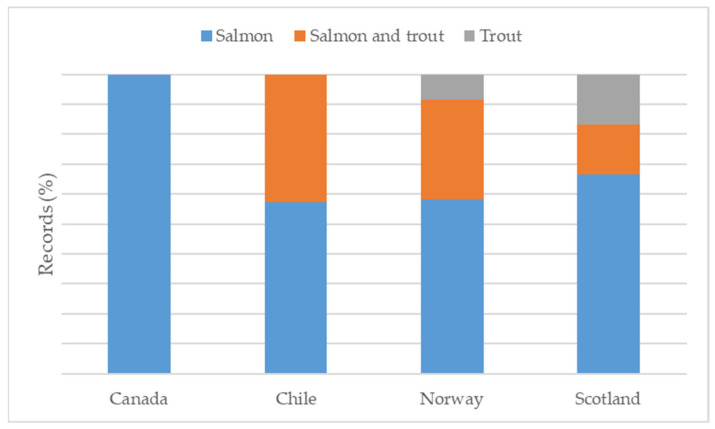
Proportions of investigated species within each country.

**Figure 4 animals-13-03525-f004:**
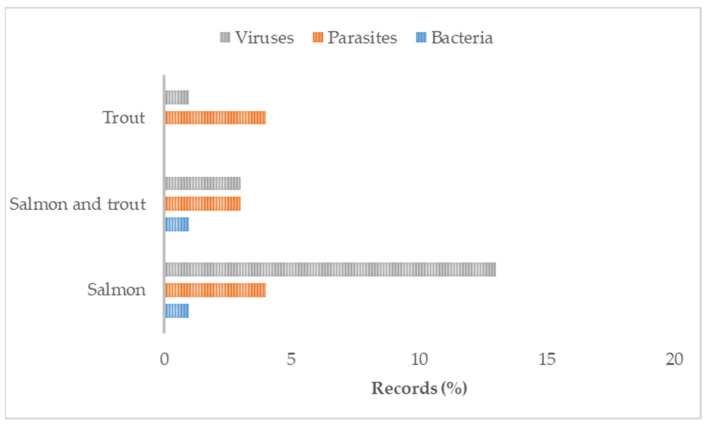
Proportions of pathological agents within species.

**Figure 5 animals-13-03525-f005:**
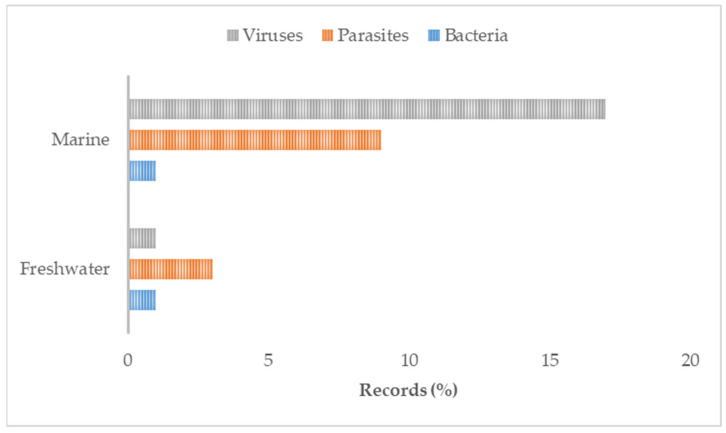
Proportions of pathological agents within environments.

**Figure 6 animals-13-03525-f006:**
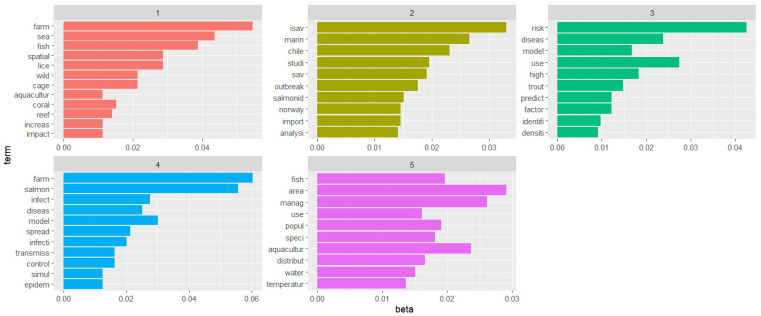
Histogram representation of the most relevant word stems within five identified research areas (beta is the probability that a stem belongs to a given research area); tentative assignment of theme titles is as follows: **1**. marine intensive fish farming and environment (red), **2**. outbreak viruses in salmon marine farming (gold), **3**. risk factors for trout diseases (green), **4**. disease epidemiology and modelling in salmon farming (blue), and **5**. fish rearing management (pink).

**Figure 7 animals-13-03525-f007:**
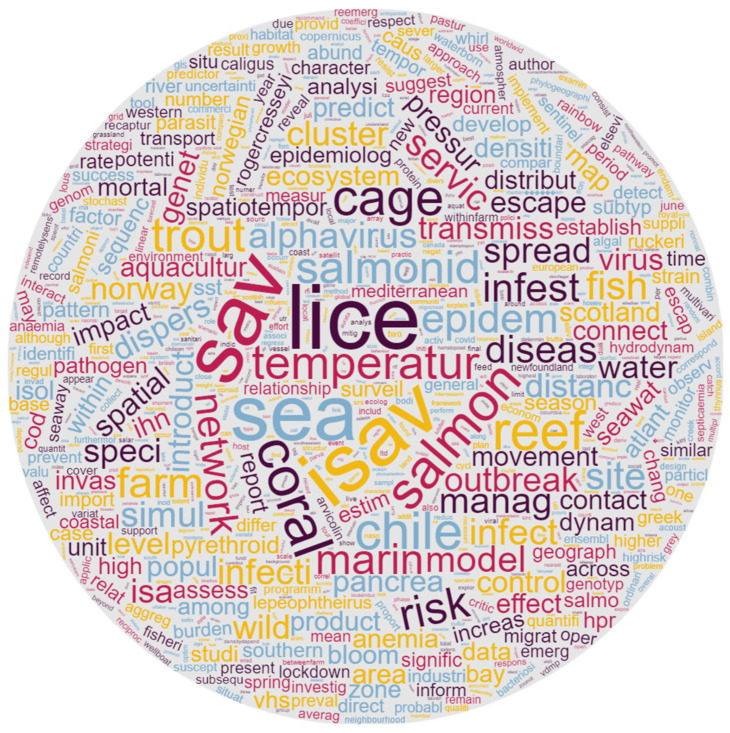
Cloud representation of the most relevant word stems in the 54 selected records. The relative importance of the terms is reflected by their size.

**Table 1 animals-13-03525-t001:** Classification of the selected references on GIS implementation for farmed fish health management according to the environment, GIS methods or techniques, and study objectives.

Environment	GIS Methods or Techniques	Study Objectives	References
Marine	Euclidean distance	Dispersal pathway; estimation of infestation pressure	[24]
	Seaway distance	Dispersal pathway; disease spread	[25]
	Euclidean distance; buffer	Simulation scenario model; risk assessment; disease management practice	[26]
		Spatiotemporal model; spatial distribution; acoustic telemetry	[19]
		Spatiotemporal analysis	[27]
		Spatial and temporal model; spatial distribution; risk assessment	[28,29]
	Seaway distance; buffer	Spatiotemporal model; stochastic model; risk map; disease spread	[10]
		Spatiotemporal analysis; risk factor assessment; surveillance support	[30]
		Simulation scenario model; stochastic model; disease management practice	[31]
		Spatiotemporal model; risk factor assessment and map	[32]
	Seaway distance; kernel density	Spatial and spatiotemporal model; spatial distribution; disease spread and management	[33,34]
	Euclidean distance; grid calculation	Spatiotemporal model; risk assessment; disease management	[35]
	Seaway distance; centroid; kernel density	Spatial clustering and dispersal pathway; estimation of infestation pressure	[36]
	Euclidean distance; nearest neighbour; grid calculation	Spatial planning; risk assessment	[37]
	Seaway distance; nearest neighbour; kernel density	Stochastic model; risk factor assessment and map; disease spread	[18]
	Seaway distance; Thiessen polygon; classification	Spatiotemporal analysis; spatial distribution; risk assessment and map	[38]
	Nearest neighbour; centroid; Haversine distance	Risk factor assessment; disease study	[39]
	Seaway distance; Euclidean distance; kernel density; raster map	Spatiotemporal analysis; risk factor assessment and map; epidemiology of infectious diseases; surveillance support	[40]
	Seaway distance; buffer; map algebra; kernel density	Spatial and temporal model; spatial distribution; risk assessment	[41]
	Seaway distance; buffer; kriging; raster map	Spatiotemporal model; risk map; disease spread	[42]
	Seaway distance; Euclidean distance; buffer; centroid; nearest neighbour	Risk factor assessment; epidemiology of infectious diseases; surveillance support	[43]
	Seaway distance; Euclidean distance; buffer; kernel density; raster map	Spatiotemporal model; risk factor assessment; epidemiology of infectious diseases	[44]
	Basic and hydrographic cartography	Spatial multi-criteria decision analysis	[45]
	None	Simulation scenario model; risk map; disease management practice	[46]
		Manipulation of satellite images; monitoring of algae bloom	[47]
		Spatial analysis; spatial distribution	[48]
Freshwater	Grid calculation; map algebra	Spatiotemporal analysis; risk assessment and map; surveillance support	[49,50]
	Euclidean distance; buffer; kernel density	Stochastic model; simulation model; spatial distribution; risk assessment and map	[51]
		Geographical distribution; risk assessment	[52]
	Generalisation; raster; map algebra	Spatial analysis; spatial distribution	[53]
	Euclidean distance; buffer; area calculation; overlay	Spatial and temporal analysis; risk assessment	[20]
	Euclidean distance; centroid; raster; map algebra	Spatial model; spatial distribution; risk assessment and map	[54]
	Euclidean distance; buffer; nearest neighbour; point-in-polygon; kriging; IDW	Spatiotemporal model; spatial distribution; risk assessment and map; epidemiology of infectious diseases; surveillance support	[12]
Marine—fjords	Buffer	Spatial distribution; risk factor assessment; phylogenetic analysis	[55]
	Seaway distance; buffer	Spatiotemporal model; simulation model; stochastic model; risk assessment and map	[9]
	Seaway distance; buffer; centroid	Spatiotemporal model; simulation model; spatial distribution; spatial dispersal model	[56]
	Seaway distance; buffer; kernel density	Simulation scenario model; risk factor assessment; disease management practice	[57]
	Euclidean distance; buffer; centroid; nearest neighbour	Spatiotemporal model; simulation model; spatial dispersal model	[58]
Marine—freshwater	Euclidean distance; buffer	Spatiotemporal analysis; risk assessment; risk map; spatial epidemiology methods to limit the risk of disease introduction and spread; surveillance support	[17]
	Euclidean distance; buffer; centroid	Spatiotemporal analysis; risk assessment and map; descriptive analysis; epidemiology of infectious diseases; spatial epidemiology methods to limit the risk of disease introduction and spread; surveillance support	[59]
	Euclidean distance; buffer; nearest neighbour	Spatiotemporal analysis; risk assessment; surveillance support	[60]
Marine—transitional	Euclidean distance; buffer	Geodata production and management; spatial distribution	[61]

## Data Availability

This work was based on previously published research; no primary data were used. The PRISMA-ScR protocol is available from the authors upon request.

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
