# Peer review of "A Scoping Review on GIS Technologies Applied to Farmed Fish Health Management"

_animals, 2023, doi:10.3390/ani13223525_

Round 1

Reviewer 1 Report

Comments and Suggestions for Authors This scoping review aims to provide an overview of the extent of the literature surrounding the current use of GIS globally, to support veterinary services in farmed fish health management in freshwater, marine and transitional environments. Indeed, scoping reviews are deemed to summarise a wide range of published evidence to understand the amplitude of a discipline, evaluate the extent and nature of research activity, and display gaps in the literature. Therefore, this explorative review study will provide a summary of original research articles that report GIS applications, databases, and functions useful for risk and predictive modelling in fish health management. This work is meaningful. 1 The language should be carefully polished. 2 The figures are confused. Authors should update them. Comments on the Quality of English Language This scoping review aims to provide an overview of the extent of the literature surrounding the current use of GIS globally, to support veterinary services in farmed fish health management in freshwater, marine and transitional environments. Indeed, scoping reviews are deemed to summarise a wide range of published evidence to understand the amplitude of a discipline, evaluate the extent and nature of research activity, and display gaps in the literature. Therefore, this explorative review study will provide a summary of original research articles that report GIS applications, databases, and functions useful for risk and predictive modelling in fish health management. This work is meaningful. 1 The language should be carefully polished. 2 The figures are confused. Authors should update them.

Author Response

Reviewer 1 (Changes highlighted in green)

This scoping review aims to provide an overview of the extent of the literature surrounding the current use of GIS globally, to support veterinary services in farmed fish health management in freshwater, marine and transitional environments. Indeed, scoping reviews are deemed to summarise a wide range of published evidence to understand the amplitude of a discipline, evaluate the extent and nature of research activity, and display gaps in the literature. Therefore, this explorative review study will provide a summary of original research articles that report GIS applications, databases, and functions useful for risk and predictive modelling in fish health management. This work is meaningful.

Authors’ reply: We thank the reviewer very much for the positive comment.

1 The language should be carefully polished.

Authors’ reply: We had the English revised by the MDPI editing service (changes are highlighted in grey).

2 The figures are confused. Authors should update them.

Authors’ reply: We apologize for the confusion that our Figure and Table arrangement might have caused. To help the reader, we cited all figures and tables in section 2.6 Synthesis of Results for the first time and added a comment at the end of it to warn that they will be then displayed consecutively in the Results (L. 239-240). Therefore, we deleted the citation of Table 1 from section 2.4 Data Charting Process and Data Items (L. 213) and moved Figure 1 to section 3.1 Selection of Sources of Evidence. In the section Results of the manuscript, we also improved the citations of Figures 3 to 5 (L. 307-309) and inverted the citations of Figures 6 and 7 (L. 320-326). Finally, Table 1 is currently vertical according to the journal’s template. It could be changed and made horizontal; however, the actual definite display of tables and figures in the final paper will be decided by the editors.

Reviewer 2 Report

Comments and Suggestions for Authors

This manuscript deals with an interesting topic. However, the following are suggested to improve this manuscript:

1. It is better to make a more comprehensive classification of topics in this field. For example, put comprehensive categories and define subsets based on them.

2. Mention the classification of modeling methods. For example, multi-criteria decision making methods, statistical methods, artificial intelligence and....

3. It is better to categorize the software used in GIS for this problem and express their frequency.

4. The challenges and limitations of existing methods in this field should be stated.

5. Suggestions for future research are given.

Comments on the Quality of English Language

Minor editing of English language required

Author Response

Reviewer 2 (changes highlighted in light blue)

This manuscript deals with an interesting topic. However, the following are suggested to improve this manuscript:

Authors’ reply: Thank you very much for the positive comment. We addressed all your comments and suggestions.

  1. It is better to make a more comprehensive classification of topics in this field. For example, put comprehensive categories and define subsets based on them.

Authors’ reply: We apologize because we could not understand whether the reviewer’s comment was about the topics described in section 2.1 Protocol and Eligibility Criteria or the research topics highlighted through the text mining analysis. Therefore, we improved both of them according to what the reviewer suggested. In particular, in section 2.1, we re-wrote the categorization and sub-categorization of the topic areas the literature records had to be related to by using bullet points (L. 141-146). As regards text mining, to avoid any misunderstanding among these and the topics in section 2.1, we changed wordings and used the terms research area/research theme to talk about the text mining analysis (L. 214-225, 236, 321, 328-330).

  1. Mention the classification of modeling methods. For example, multi-criteria decision making methods, statistical methods, artificial intelligence and....

Authors’ reply: Such classification, as suggested by the reviewer, is already available in the supplementary material (Table S1). However, we now included a more comprehensive description of the information collected from the selected records (L. 193-211). Moreover, a relevant description of them is included in the section 3.2 Characteristics and Results of Sources of Evidence.

  1. It is better to categorize the software used in GIS for this problem and express their frequency.

Authors’ reply: We thank the reviewer for this comment. However, we think a detailed categorization of the software used in GIS is already available in the manuscript, together with the frequency of use of the most popular software (ArcGIS and R) (L. 284-289). A further categorization would be too articulated and not meaningful for the purposes of the present scoping review.

  1. The challenges and limitations of existing methods in this field should be stated.

Authors’ reply: We thank the reviewer for arising this point. However, as reported by Munn et al. (BMC Medical Research Methodology, 2018, 18:143), scoping reviews do not aim to produce a critically appraised and synthesised result/answer to a particular question, and rather aim to provide an overview or map of the evidence. Nevertheless, we added some statements regarding current challenges and limitations and suggestions for future research (L. 455-461, 493-501).

  1. Suggestions for future research are given.

Authors’ reply: As answered to the previous comment, despite out of scoping review’s purposes, we added some statements regarding current challenges and limitations and suggestions for future research (L. 455-461, 493-501).

Minor editing of English language required

Authors’ reply: We had the English revised by the MDPI editing service (changes are highlighted in grey).

Reviewer 3 Report

Comments and Suggestions for Authors

I have thoroughly reviewed your manuscript titled "A Scoping Review on GIS Technologies Applied to Farmed Fish Health Management" and would like to provide some feedback for your consideration.

  1. Structure and Clarity: While I understand that the manuscript is categorized as a review, the inclusion of both a 'Simple Summary' and an 'Abstract' can be redundant. It might be beneficial to streamline this information to avoid repetition and enhance clarity.

  2. Introduction: The introduction, although providing a general backdrop, could delve deeper into the specific nuances and challenges in the realm of GIS for farmed fish health management. Highlighting the gaps or controversies in current literature can provide readers with a clearer understanding of the review's motivation.

  3. Methodology: While the methodology for paper selection follows a standard approach, elucidating any unique criteria or considerations taken during the selection process might offer readers a clearer perspective on the scope of your review.

  4. Results & Discussion: The review provides a comprehensive overview of existing literature. However, emphasizing trends, gaps, or particularly novel findings within the selected papers could make the review more impactful. Offering critical insights or suggestions for future research based on your findings can further enhance its value.

  5. Overall Contribution: Given the nature of the review, it's essential to clarify its unique contribution to the field. Is it the first of its kind in this specific niche? Does it offer a different perspective from existing reviews? Highlighting this can bolster the review's significance.

In conclusion, the topic you've chosen is undoubtedly relevant. A few refinements in terms of depth, critical analysis, and emphasis on the review's unique contribution can elevate its impact and usefulness to the readership.

Author Response

Reviewer 4 (changes highlighted in purple)

I have thoroughly reviewed your manuscript titled "A Scoping Review on GIS Technologies Applied to Farmed Fish Health Management" and would like to provide some feedback for your consideration.

Authors’ reply: Thank you very much for your positive revision and useful considerations. Your considerations have been addressed.

Structure and Clarity: While I understand that the manuscript is categorized as a review, the inclusion of both a 'Simple Summary' and an 'Abstract' can be redundant. It might be beneficial to streamline this information to avoid repetition and enhance clarity.

Authors’ reply: We do agree on the reviewer’s comment, however, we followed the template provided by the journal for submissions, which includes both “simple summary” and “abstract”. It was requested to include both of them in the submitted manuscript.

Introduction: The introduction, although providing a general backdrop, could delve deeper into the specific nuances and challenges in the realm of GIS for farmed fish health management. Highlighting the gaps or controversies in current literature can provide readers with a clearer understanding of the review's motivation.

Authors’ reply: So as to providing a stronger motivation for our review, we improved the Introduction as suggested, which is by describing the current challenges in the use of GIS for farmed fish health management and gaps in the literature (L. 103-108).

Methodology: While the methodology for paper selection follows a standard approach, elucidating any unique criteria or considerations taken during the selection process might offer readers a clearer perspective on the scope of your review.

Authors’ reply: We thank the reviewer for the suggestion and agree on the fact that a precise description of our selection criteria makes the perspective on the scope of this review clearer for the reader. While developing and describing our literature search workflow and selection of final records we were as strict and precise as possible (see sections 2.2 and 2.3 and Figure 1); so as to allow anyone else to repeat the same scoping review procedure. However, as this is a scoping review, the authors’ aim is not to investigate a very specific and unique question (see Munn et al., BMC Medical Research Methodology, 2018, 18:143), but rather to provide an overview or map of the evidence on a topic. For this reason, our search workflow was based on a “net” that was as strict as possible so as to be meaningful for our scope but also wide so we could be sure we included all the meaningful records (original research papers) in the first literature harvest, and eliminate what was, in fact, not relevant through the following selection process. Having a wider initial “net” was also necessary because, as explained after your following comment, there is very little literature work on this specific research niche and, therefore, the risk of losing potentially eligible material since the very beginning was very high.

Results & Discussion: The review provides a comprehensive overview of existing literature. However, emphasizing trends, gaps, or particularly novel findings within the selected papers could make the review more impactful. Offering critical insights or suggestions for future research based on your findings can further enhance its value.

Authors’ reply: We thank the reviewer for arising this point. Despite the fact that, as reported by Munn et al. (BMC Medical Research Methodology, 2018, 18:143), scoping reviews’ key feature is not about producing implications for practice in terms of concrete guidance, we nonetheless have already delved into the description of critical insights or suggestions for future research, together with the highlighting of trends and gaps in the literature. Nevertheless, we added some new statements regarding current challenges and limitations and suggestions for future research (L. 455-461, 493-501).

Overall Contribution: Given the nature of the review, it's essential to clarify its unique contribution to the field. Is it the first of its kind in this specific niche? Does it offer a different perspective from existing reviews? Highlighting this can bolster the review's significance.

Authors’ reply: As suggested by the reviewer, while also keeping in mind what are the objectives of scoping reviews, we clarified the unique contribution and novelty provided by this manuscript so as to highlight its significance in the field (L. 127-130, 440-442).

In conclusion, the topic you've chosen is undoubtedly relevant. A few refinements in terms of depth, critical analysis, and emphasis on the review's unique contribution can elevate its impact and usefulness to the readership.

Authors’ reply: We thank the reviewer for the positive comment and useful suggestions. Your comments and suggestions have been addressed and followed.

Round 2

Reviewer 2 Report

Comments and Suggestions for Authors

Accept in present form.

Author Response

Reviewer 2

The authors are glad that you are satisfied with our replies to all your comments and suggestions.

Reviewer 3 Report

Comments and Suggestions for Authors

Dear Authors,

Upon reviewing the revised version of your manuscript "A Scoping Review on GIS Technologies Applied to Farmed Fish Health Management," I have noted the following issues that require attention:

  1. Inconsistency in Selection Criteria: There is an inconsistency within the manuscript concerning the inclusion criteria for the reviewed studies. Specifically, the text from lines 138 to 146 outlines a broad set of topics for study selection that does not necessarily require a focus on GIS. This is contradictory to the search query detailed between lines 158 and 161, which clearly stipulates the need for GIS-related keywords. This inconsistency could lead to confusion about the focus and conclusions of the review. Clarification on this matter is essential to ensure the integrity of the review process and the relevance of the included studies to the research question.

  2. Exclusion of Non-English Papers: The manuscript excludes non-English papers from the review. Given the advancement of translation technologies, the exclusion of studies based on language constraints could limit the scope and comprehensiveness of the review. I recommend providing a rationale for this exclusion and considering the potential benefits of including relevant research in other languages to enrich the review.

  3. Depth of Analysis: The current analysis appears to be predominantly syntactic, focusing on keyword occurrence rather than the underlying meaning and context of the content. A more semantic approach to data analysis could offer a deeper understanding of the application and impact of GIS technologies in fish health management. I suggest enhancing the depth of the analysis to capture more nuanced insights from the literature.

I hope these comments are helpful in strengthening the manuscript for publication.

Author Response

Reviewer 3 (changes highlighted in purple)

Dear Authors,

Upon reviewing the revised version of your manuscript "A Scoping Review on GIS Technologies Applied to Farmed Fish Health Management," I have noted the following issues that require attention:

  1. Inconsistency in Selection Criteria: There is an inconsistency within the manuscript concerning the inclusion criteria for the reviewed studies. Specifically, the text from lines 138 to 146 outlines a broad set of topics for study selection that does not necessarily require a focus on GIS. This is contradictory to the search query detailed between lines 158 and 161, which clearly stipulates the need for GIS-related keywords. This inconsistency could lead to confusion about the focus and conclusions of the review. Clarification on this matter is essential to ensure the integrity of the review process and the relevance of the included studies to the research question.

Authors reply - The authors thank the reviewer for arising this point, and we agree on the fact that the description of the overall topics in section 2.1 Protocol and Eligibility Criteria could generate an inconsistency. We apologize for this mistake, and we changed the related sentence (L. 139-140).

  1. Exclusion of Non-English Papers: The manuscript excludes non-English papers from the review. Given the advancement of translation technologies, the exclusion of studies based on language constraints could limit the scope and comprehensiveness of the review. I recommend providing a rationale for this exclusion and considering the potential benefits of including relevant research in other languages to enrich the review.

Authors reply - The authors thank the Reviewer for this consideration, however we only partially agree for the following reasons. Firstly, there were only very few non-English papers related to the objectives of this SR and their exclusion did not cause the loss of any relevant meaningful material for the manuscript. Furthermore, despite being true that today’s advanced translation technologies allow good translations from many languages into English, when it comes to scientific literature, the vast majority, if not all, of it is requested to be in English to be published in peer-reviewed journals, which are the source elected for this scoping review. Thirdly, the innovative text mining (TM) literature search processing (see the generated research areas of Figure 6, and word stem cloud of Figure 7), that we applied together with the more descriptive analysis typical of reviews, implied the election of only one language used in all the analysed text corpus.

  1. Depth of Analysis: The current analysis appears to be predominantly syntactic, focusing on keyword occurrence rather than the underlying meaning and context of the content. A more semantic approach to data analysis could offer a deeper understanding of the application and impact of GIS technologies in fish health management. I suggest enhancing the depth of the analysis to capture more nuanced insights from the literature.

Authors reply – As already replied to the Reviewer in Round 1 regarding this suggestion, scoping reviews’ key feature is not about producing implications for practice in terms of concrete guidance (see Munn et al., BMC Medical Research Methodology, 2018, 18:143). However, we have added new statements regarding further insights coming from the current and future potential applications and impact of GIS technologies in fish health management (L. 429-439; 451-460; 488-501) and a new reference (n. 71).

I hope these comments are helpful in strengthening the manuscript for publication.

Authors reply – As for your first revision, also the second one was helpful to improve the quality of our manuscript. Thank you for your help.

Round 3

Reviewer 3 Report

Comments and Suggestions for Authors

Thanks for update and explanation.